# BiGrad: Differentiating through Bilevel Optimization Programming

## Francesco Alesiani

NEC Laboratories Europe
Heidelberg, Germany, DE
francesco.alesiani@neclab.eu

## Abstract

Integrating mathematical programming, and in particular Bilevel Optimization Programming, within deep learning architectures has vast applications in various domains from machine learning to engineering. Bilevel programming is able to capture complex interactions when two actors have conflicting objectives. Previous approaches only consider single-level programming. In this paper, we thus propose Differentiating through Bilevel Optimization Programming (BiGrad) as approach for end-to-end learning of models that use Bilevel Programming as a layer. BiGrad has wide applicability and it can be used in modern machine learning frameworks. We focus on two classes of Bilevel Programming: continuous and combinatorial optimization problems. The framework extends existing approaches of single level optimization programming. We describe a class of gradient estimators for the combinatorial case which reduces the requirements in term of computation complexity; for the continuous variables case the gradient computation takes advantage of push-back approach (i.e. vector-jacobian product) for an efficient implementation. Experiments suggest that the proposed approach successfully extends existing single level approaches to Bilevel Programming.

## 1 Introduction

Neural networks provide unprecedented improvements in perception tasks, however, they struggle to learn basic logic operations (Garcez et al. 2015) or relationships. When modelling complex systems, for example decision systems, it is not only beneficial to integrate optimization components into larger differentiable system, but also to use general purpose solvers (e.g. for Integer Linear Programming or Nonlinear Programming (Bertsekas 1997; Boyd and Vandenberghe 2004)) and problem specific implementation, to discover the governing discrete or continuous relationships. Recent approaches propose thus differentiable layers that incorporate either quadratic (Amos and Kolter 2017), convex (Agrawal et al. 2019a), cone (Agrawal et al. 2019b), equilibrium (Bai, Kolter, and Koltun 2019), SAT (Wang et al. 2019) or combinatorial (Pogančić et al. 2019; Mandi and Guns 2020; Berthet et al. 2020) programs. Use of optimization programming as layer of differentiable systems, requires to compute the gradients through these layers, which

is either specific to the optimization problem or zero almost everywhere, when dealing with discrete variables. Proposed gradient estimates either relax the combinatorial problem (Mandi and Guns 2020), or perturb the input variables (Berthet et al. 2020; Domke 2010) or linearly approximate the loss function (Pogančić et al. 2019).

These approaches though, do now allow to directly express models with conflicting objectives, for example in structural learning (Elsken, Metzen, and Hutter 2019) or adversarial system (Goodfellow et al. 2014). We thus consider the use of bilevel optimization programming as a layer. Bilevel Optimization Program (Kleinert et al. 2021; Colson, Marcotte, and Savard 2007; Dempe 2018; Stackelberg et al. 1952), also known as generalization of Stackelberg Games, is the extension of single-level optimization program, where the solution of one optimization problem (i.e. the outer problem) depends on the solution of another optimization problem (i.e. the inner problem). This class of problems can model interactions between two actors[1], where the action of the first depends on the knowledge of the counter-action of the second. Bilevel Programming finds application in various domains, as in Electricity networks, Economics, Environmental policy, Chemical plant, defence and planning (Dempe 2018; Sinha, Malo, and Deb 2017). In general, Bilevel programs are NP-hard (Sinha, Malo, and Deb 2017), they require specialized solvers and it is not clear how to extend previous approaches, since standard chain rule is not directly applicable.

By modelling the bilevel optimization problem as an implicit layer (Bai, Kolter, and Koltun 2019), we consider the more general case where 1) the solution of the bilevel problem is computed separately by a bilevel solver; thus leveraging on powerfully solver developed over various decades (Kleinert et al. 2021); and 2) the computation of the gradient is more efficient, since we do not have to propagate gradient through the solver. We thus propose Differentiating through Bilevel Optimization Programming (BiGrad):

- BiGrad comprises of forward pass, where existing solvers can be used, and backward pass, where BiGrad estimates gradient for both continuous and combinatorial problems based on sensitivity analysis;
- we show how the proposed gradient estimators relate

---

[1] In the following section we provide concrete example of applications.

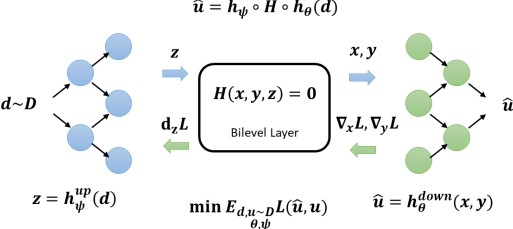

$$\hat{u} = h_\psi \circ H \circ h_\theta(d)$$

$H(x, y, z) = 0$

Bilevel Layer

$d \sim D$

$z$

$\mathbf{d}_z L$

$x, y$

$\nabla_x L, \nabla_y L$

$\hat{u}$

$z = h_\psi^{up}(d)$      $\min_{\theta, \psi} E_{d, u \sim D} L(\hat{u}, u)$      $\hat{u} = h_\theta^{down}(x, y)$

Figure 1: The Forward and backward passes of a Bilevel Programming layer: the larger system has input $d$ and output $u = h_\psi \circ H \circ h_\theta(d)$; the bilevel layer has input $z$ and output $x, y$, which are solutions of a Bilevel optimization problem represented by the implicit function $H(x, y, z) = 0$.

to the single-level analogous and that the proposed approach is beneficial in both continuous and discrete cases.

**Examples of Bilevel Optimization Problems**

**Physical System with control sub-system example** Bilevel Programming is to model the interaction of a dynamical system ($x$) and its control sub-system ($y$), as for example an industrial plant or a physical process. The control sub-system changes based on the state of the underlying dynamical system, which itself solves a physics constraint optimization problem (Raissi, Perdikaris, and Karniadakis 2019; de Avila Belbute-Peres et al. 2018).

**Interdiction problem example** Two actors discrete Interdiction problems (Fischetti et al. 2019), where one actor ($x$) tries to interdict the actions of another actors ($y$) under budget constraints, arise in various areas, from marketing, protecting critical infrastructure, preventing drug smuggling to hinder nuclear weapon proliferation.

**Min-max problem example** Min-max problems are used to model robust optimization problems (Ben-Tal, El Ghaoui, and Nemirovski 2009), where a second variable represents the environment and is constrained to an uncertain set that captures the unknown variability of the environment.

**Adversarial attack in Machine Learning** Bilevel probram is used the represents the interaction between a machine learning model ($y$) and a potential attacker ($x$) (Goldblum, Fowl, and Goldstein 2019) and is used to increase the resilience to intentional or unintended adversarial attacks.

## 2 Differentiable Bilevel Optimization Layer

We model the Bilevel Optimization Program as an Implicit Layer (Bai, Kolter, and Koltun 2019), i.e. as the solution of an implicit equation $H(x, y, z) = 0$, in order to derive the gradient using the implicit function theorem, where $z$ is given and represents the parameters of our system we want to estimate, and $x, y$ are output variables (Fig.1). We also assume we have access[2] to a solver $(x, y) = \text{Solve}_H(z)$. The bilevel Optimization Program is then used a layer of a differentiable system, whose input is $d$ and output is given by

$u = h_\psi \circ \text{Solve}_H \circ h_\theta(d) = h_{\psi,\theta}(d)$, where $\circ$ is the function composition operator. We want to learn the parameters $\psi, \theta$ of the function $h_{\psi,\theta}(d)$ that minimize the loss function $L(h_{\psi,\theta}(d), u)$, using the training data $D^{\text{tr}} = \{(d, u)_{i=1}^{N^{\text{tr}}}\}$. In order to be able to perform the end-to-end training, we need to back-propagate the gradient of the Bilevel Optimization Program Layer, which can not be accomplish only using chain rule.

### 2.1 Continuous Bilevel Programming

We now present the definition of the continous Bilevel Optimization problem, which comprises of two non-linear function $f, g$, as

$$\min_{x \in X} f(x, y, z) \qquad y \in \arg\min_{y \in Y} g(x, y, z) \qquad (1)$$

where the left part problem is called *outer optimization problem* and resolves for the variable $x \in X$, with $X = \mathbb{R}^n$. The right problem is called the *inner optimization problem* and solves for the variable $y \in Y$, with $Y = \mathbb{R}^m$. The variable $z \in \mathbb{R}^p$ is the input variable and is a parameter for the bilevel problem. Min-max is special case of Bilevel optimization problem $\min_{y \in Y} \max_{x \in X} g(x, y, z)$, where the minimization functions are equal and opposite in sign.

### 2.2 Combinatorial Bilevel Programming

When the variables are discrete, we restrict the objective functions to be multi-linear (Greub 1967). Various important combinatorial problems are linear in discrete variables (e.g. VRP, TSP, SAT [3]), one example form is the following

$$\min_{x \in X} \langle z, x \rangle_A + \langle y, x \rangle_B, \ \ y \in \arg\min_{y \in Y} \langle w, y \rangle_C + \langle x, y \rangle_D \qquad (2)$$

The variables $x, y$ have domains in $x \in X, y \in Y$, where $X, Y$ are convex polytopes that are constructed from a set of distinct points $\mathcal{X} \subset \mathbb{R}^n, \mathcal{Y} \subset \mathbb{R}^m$, as their convex hull. The outer and inner problems are Integer Linear Programs (ILPs). The multi-linear operator is represented by the inner product $\langle x, y \rangle_A = x^T A y$. We only consider the case where we have separate parameters for the outer and inner problems, $z \in \mathbb{R}^p$ and $w \in \mathbb{R}^q$.

## 3 BiGrad: Gradient estimation

Even if the discrete and continuous variable cases share a similar structure, the approach is different when evaluating the gradients. We can identify the following common basic steps (Alg.1):

1. In the forward pass, solve the combinatorial or continuous Bilevel Optimisation problem as defined in Eq.1(or Eq.2) using existing solver;
2. During the backward pass, compute the gradient $\mathrm{d}_z L$ (and $\mathrm{d}_w L$) using the suggested gradients (Sec.3.1 and Sec.3.2) starting from the gradients on the output variables $\nabla_x L$ and $\nabla_y L$.

---

[2] Finding the solution of the bi-level problem is not in the scope of this work.

[3] Vehicle Routing Problem, Boolean satisfiability problem.

---

**Algorithm 1: BiGrad Layer: Bilevel Optimization Programming Layer using BiGrad**

---

1. **Input**: Training sample $(\tilde{d}, \tilde{u})$
2. **Forward Pass**:
   (a) Compute $(x, y) \in \{x, y : H(x, y, z) = 0\}$ using Bilevel Solver: $(x, y) \in \text{Solve}_H(z)$
   (b) Compute the loss function $L(h_\psi \circ H \circ h_\theta(\tilde{d}), \tilde{u})$,
   (c) Save $(x, y, z)$ for the backward pass
3. **Backward Pass**:
   (a) update the parameter of the downstream layers $\psi$ using back-propagation
   (b) For the continuous variable case, compute based on Theorem 2 around the current solution $(x, y, z)$, without solving the Bilevel Problem
   (c) For the discrete variable case, use the gradient estimates of Theorem 3 or Section 3.2 (e.g. Eq.11 or Eq.12) by solving, when needed, for the two separate problems
   (d) Back-propagate the estimated gradient to the downstream parameters $\theta$

---

## 3.1 Continuous Optimization

To evaluate the gradient of the variables $z$ versus the loss function $L$, we need to propagate the gradients of the two output variables $x, y$ through the two optimization problems. We can use the implicit function theorem to approximate locally the function $z \to (x, y)$. We have thus the following main results[4].

**Theorem 1.** *Consider the bilevel problem of Eq.1, we can build the following set of equations that represent the equivalent problem around a given solution $x^*, y^*, z^*$:*

$$F(x, y, z) = 0 \qquad G(x, y, z) = 0 \qquad (3)$$

*where*

$$F(x, y, z) = \nabla_x f - \nabla_y f \nabla_y G \nabla_x G, \quad G(x, y, z) = \nabla_y g \tag{4}$$

*where we used the short notation $f = f(x, y, z), g = g(x, y, z), F = F(x, y, z), G = G(x, y, z)$*

**Theorem 2.** *Consider the problem defined in Eq.1, then the total gradient of the parameter $z$ w.r.t. the loss function $L(x, y, z)$ is computed from the partial gradients $\nabla_x L, \nabla_y L, \nabla_z L$ as*

$$\mathrm{d}_z L = \nabla_z L - |\nabla_x L \quad \nabla_y L| \begin{vmatrix} \nabla_x F & \nabla_y F \\ \nabla_x G & \nabla_y G \end{vmatrix}^{-1} \begin{vmatrix} \nabla_z F \\ \nabla_z G \end{vmatrix} \tag{5}$$

The implicit layer is thus defined by the two conditions $F(x, y, z) = 0$ and $G(x, y, z) = 0$. We notice that Eq.5 can be solved without explicitly computing the Jacobian matrices and inverting the system, but adopting the Vector-Jacobian product approach we can proceed from left to right to evaluate $\mathrm{d}_z L$. In the following section we describe how

---
[4] Proofs are in the Supplementary Material

affine equality constraints and nonlinear inequality can be used when modelling $f, g$. We also notice that the solution of Eq.5 does not require to solve the original problem, but only to apply matrix-vector products, i.e. linear algebra, and the evaluation of the gradient that can be computed using automatic differentiation.

**Linear Equality constraints** To extend the model of Eq.1 to include linear equality constraints of the form $Ax = b$ and $By = c$ on the outer and inner problem variables, we use the following change of variables

$$x \to x_0 + A^\perp x, \ y \to y_0 + B^\perp y, \tag{6}$$

where $A^\perp, B^\perp$ are the orthogonal space of $A$ and $B$, i.e. $AA^\perp = 0, BB^\perp = 0$, and $x_0, y_0$ are one solution of the equations, i.e. $Ax_0 = b, By_0 = c$.

**Non-linear Inequality constraints** Similarly, to extend the model of Eq.1 when we have non-linear inequality constraints, we use the barrier method approach (Boyd and Vandenberghe 2004), where the variable is penalized with a logarithmic function to violate the constraints. Specifically, let us consider the case where $f_i, g_i$ are inequality constraint functions, i.e. $f_i < 0, g_i < 0$, for the outer and inner problems. We then define new functions

$$f \to tf - \sum_{i=1}^{k_x} \ln(-f_i), \ g \to tg - \sum_{i=1}^{k_y} \ln(-g_i). \tag{7}$$

where $t$ is a variable parameter, which depends on the violation of the constraints. The closer the solution is to violate the constraints, the larger the value of $t$ is.

**Bilevel Cone programming** We show here how Theorem.2 can be applied to bi-level cone programming extending single-level cone programming results (Agrawal et al. 2019b), where we can use efficient solvers for cone programs to compute a solution of the bilevel problem (Ouattara and Aswani 2018)

$$\min_x c^T x + (Cy)^T x$$
$$\text{s.t. } Ax + z + R(y)(x - r) = b, \ s \in \mathcal{K} \tag{8a}$$
$$y \in \arg\min_y d^T y + (Dx)^T y$$
$$\text{s.t. } By + u + P(x)(y - p) = f, \ u \in \mathcal{K} \tag{8b}$$

In this bilevel cone programming, the inner and outer problem are both cone programs, where $R(y), P(x)$ represents a linear transformation, while $C, r, D, p$ are new parameters of the problem, while $\mathcal{K}$ is the conic domain of the variables. In the hypothesis that a local minima of Eq.8 exists, we can use an interior point method to find such point. To compute the bilevel gradient, we then use the residual maps (Busseti, Moursi, and Boyd 2019) of the outer and inner problems. Indeed, we can then apply Theorem 2, where $F = N_1(x, Q, y)$ and $G = N_2(y, Q, x)$ are the normalized residual maps defined in (Busseti, Moursi, and Boyd 2019; Agrawal et al. 2019a) of the outer and inner problems.

## 3.2 Combinatorial Optimization

When we consider discrete variables, the gradient is zero almost everywhere. We thus need to resort to estimate gradients. For the bilevel problem with discrete variables of Eq.2, when the solution of the bilevel problem exists and its solution is given (Kleinert et al. 2021), Thm.3 gives the gradients of the loss function with respect to the input parameters.

**Theorem 3.** *Given the Eq.2 problem, the partial variation of a cost function $L(x, y, z, w)$ on the input parameters has the following form:*

$$d_z L = \nabla_z L + [\nabla_x L + \nabla_y L \nabla_x y] \nabla_z x \quad (9a)$$
$$d_w L = \nabla_w L + [\nabla_x L \nabla_y x + \nabla_y L] \nabla_w y \quad (9b)$$

The $\nabla_x y, \nabla_y x$ terms capture the interaction between outer and inner problems. We could estimate the gradients in Thm.3 using the perturbation approach suggested in (Berthet et al. 2020), which estimate the gradient as the expected value of the gradient of the problem after perturbing the input variable, but, similar to REINFORCE (Williams 1992), this introduces large variance. While it is possible to reduce variance in some cases (Grathwohl et al. 2017) with the use of additional trainable functions, we consider alternative approaches as described in the following.

**Differentiation of blackbox combinatorial solvers** (Pogančić et al. 2019) propose a way to propagate the gradient through a single level combinatorial solver, where $\nabla_z L \approx \frac{1}{\tau}[x(z + \tau \nabla_x L) - x(z)]$ when $x(z) = \arg\max_{x \in X} \langle x, z \rangle$. We thus propose to compute the variation on the input variables from the two separate problems of the Bilevel Problem:

$$\nabla_z L \approx 1/\tau[x(z + \tau A \nabla_x L, y) - x(z, y)] \quad (10a)$$
$$\nabla_w L \approx 1/\tau[y(w + \tau C \nabla_y L, x) - y(w, x)] \quad (10b)$$

or alternatively, if we have only access to the Bilevel solver and not to the separate ILP solvers, we can express

$$\nabla_{z,w} L \approx 1/\tau[s(v + \tau E \nabla_{x,y} L) - s(v)] \quad (11)$$

where $x(z, y)$ and $y(w, x)$ represent the solutions of the two problems separately, $s(v) = (z, w) \rightarrow (x, y)$ the complete solution to the Bilevel Problem, $\tau \rightarrow 0$ is a hyper-parameter and $E = \begin{bmatrix} A & 0 \\ 0 & C \end{bmatrix}$. This form is more convenient than Eq.9, since it does not require to compute the cross terms, ignoring thus the interaction of the two levels.

**Straight-Through gradient** In estimating the input variables $z, w$ of our model, we may not be interested in the interaction between the two variable $x, y$. Let us consider, for example, the squared $\ell_2$ loss function defined over the output variables

$$L^2(x, y) = L^2(x) + L^2(y)$$

where $L^2(x) = \frac{1}{2}\|x - x^*\|_2^2$ and $x^*$ is the true value. The loss is non zero only when the two vectors disagree, and with integer variables, it counts the difference squared or, in case of the binary variables, it counts the number of differences. If we compute $\nabla_x L^2(x) = (x - x^*)$ in the binary case, we

have that $\nabla_{x_i} L^2(x) = +1$ if $x_i^* = 0 \wedge x_i = 1$, $\nabla_{x_i} L^2(x) = -1$ if $x_i^* = 1 \wedge x_i = 0$, and 0 otherwise. This information can be directly used to update the $z_i$ variable in the linear term $\langle z, x \rangle$, thus we can estimate the gradients of the input variables as $\nabla_{z_i} L^2 = -\lambda \nabla_{x_i} L^2$ and $\nabla_{w_i} L^2 = -\lambda \nabla_{y_i} L^2$, with some weight $\lambda > 0$. The intuition is that, the weight $z_i$ associated with the variable $x_i$ is increased, when the value of the variable $x_i$ reduces. In the general multilinear case we have additional multiplicative terms. Following this intuition (see Sec.A.3), we thus use as an estimate of the gradient of the variables

$$\nabla_z L = -A\nabla_x L \qquad \nabla_w L = -C\nabla_y L \quad (12)$$

This is equivalent in Eq.2 where $\nabla_z x = \nabla_w y = -I$ and $\nabla_y x = 0$, thus $\nabla_x y = 0$. This update is also equivalent to Eq.10, without the soluton computation. The advantage of this form is that it does not requires to solve for an additional solution in the backward pass. For the single level problem, gradient has the same form of the Straight-Through gradient proposed by (Bengio, Léonard, and Courville 2013), with surrogate gradient $\nabla_z x = -I$.

## 4 Related Work

**Bilevel Programming in machine learning** Various papers model machine learning problem as Bilevel problems, for example in Hyper-parameter Optimization (MacKay et al. 2019; Franceschi et al. 2018), Meta-Feature Learning (Li and Malik 2016), Meta-Initialization Learning (Rajeswaran et al. 2019), Neural Architecture Search (Liu, Simonyan, and Yang 2018), Adversarial Learning (Li et al. 2019), Deep Reinforcement Learning (Vahdat et al. 2020) and Multi-Task Learning (Alesiani et al. 2020). In these works the main focus is to compute the solution of the bilevel optimization problems. In (MacKay et al. 2019; Lorraine and Duvenaud 2018), the best response function is modeled as a neural network and the solution is found using iterative minimization, without attempting to estimate the complete gradient. Many bilevel approaches rely on the use of the implicit function to compute the hyper-gradient (Sec. 3.5 of (Colson, Marcotte, and Savard 2007)), but do not use bilevel as layer.

**Quadratic, Cone and Convex single-level Programming** Various works have addressed the problem of differentiate through quadratic, convex or cone programming (Amos 2019; Amos and Kolter 2017; Agrawal et al. 2019b,a). In these approaches the optimization layer is modelled as an implicit layer and for the cone/convex case the normalized residual map is used to propagate the gradients. Contrary to our approach, these work only address single level problems. These approaches do not consider combinatorial optimization.

**Implicit layer Networks** While classical deep neural neural networks perform a single pass through the network at inference time, a new class of systems performs inference by solving an optimization problem. Example of this are Deep Equilibrium Network (DEQ) (Bai, Kolter, and Koltun 2019) and NeuroIODE (NODE) (Chen et al. 2018). Similar to our

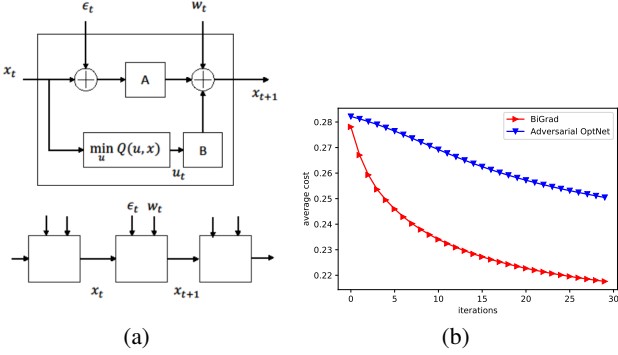

(a)                              (b)

Figure 2: (a) Visualization of the Optimal Control Learning network, where a disturbance $\epsilon_t$ is injected based on the control signal $u_t$. (b) Comparison of the training performance for $N = 2$, $T = 20$ and epochs=10 of the BiGrad and the Adversarial version of the OptNet (Amos and Kolter 2017).

approach, the gradient is computed based on sensitivity analysis of the current solution. These methods only consider continuous optimization.

**Combinatorial optimization** Various papers estimate gradients of single-level combinatorial problems using relaxation. (Wilder, Dilkina, and Tambe 2019; Elmachtoub and Grigas 2017; Ferber et al. 2020; Mandi and Guns 2020) for example use $\ell_1, \ell_2$ or log barrier to relax the Integer Linear Programming (ILP) problem. Once relaxed the problem is solved using standard methods for continuous variable optimization. An alternative approach is suggested in other papers. For example in (Pogančić et al. 2019) the loss function is approximated with a linear function and this leads to an estimate of the gradient of the input variable similar to the implicit differentiation by perturbation form (Domke 2010). (Berthet et al. 2020) is another approach that uses also perturbation and change of variables to estimate the gradient in a ILP problem. SatNet (Wang et al. 2019) solves MAXSAT problems by solving a continuous semidefinite program (SDP) relaxation of the original problem. These works only consider single-level problems.

**Discrete latent variables** Discrete random variables provide an effective way to model multi-modal distributions over discrete values, which can be used in various machine learning problems, e.g. in language models (Yang et al. 2017) or for conditional computation (Bengio, Léonard, and Courville 2013). Gradients of discrete distribution are not mathematical defined, thus, in order to use gradient based method, gradient estimations have been proposed. A class of methods is based on Gumbel-Softmax estimator (Jang, Gu, and Poole 2016; Maddison, Mnih, and Teh 2016; Paulus, Maddison, and Krause 2021).

## 5 Experiments

We evaluate BiGrad with continuous and combinatorial problems to shows that improves over single-level approaches. In the first experiment we compare the use of BiGrad versus the use of the implicit layer proposed in (Amos

Table 1: Optimal Control Average Cost; Bilevel approach improves (lower cost) over two-step approach, because is able to better capture the interaction between noise and control dynamics.

|  | LQR | OptNet | Bilevel |
|---|---|---|---|
| Adversarial (10 steps) | 2.736 | 0.2722 | **0.2379** |
| (30 steps) | - | 0.2511 | **0.2181** |

and Kolter 2017) for the design of Optimal Control with adversarial noise. In the second part, after experimenting with adversarial attack, we explore the performance of BiGrad with two combinatorial problems with Interdiction, where we adapted the experimental setup proposed in (Pogančić et al. 2019). In these latter experiments, we compare the formulation in Eq.11 (denoted by Bigrad(BB)) and the formulation of Eq.12 (denoted by Bigrad(PT)). In addition we compare with the single level BB-1 from (Pogančić et al. 2019) and single level straight-through (Bengio, Léonard, and Courville 2013; Paulus, Maddison, and Krause 2021), with the surrogate gradient $\nabla_z x = -I$, (PT-1) gradient estimations. We compare against Supervised learning (SL), which ignores the underlay structure of the problem and directly predicts the solution of the bilevel problem.

### 5.1 Optimal Control with adversarial disturbance

We consider the design of a robust stochastic control for a Dynamical System (Agrawal et al. 2019b). The problem is to find a feedback function $u = \phi(x)$ that minimizes

$$\min_{\phi} \mathbb{E} \frac{1}{T} \sum_{t=0}^{T} \|x_t\|^2 + \|\phi(x_t)\|^2 \qquad (13a)$$

$$\text{s.t. } x_{t+1} = Ax_t + B\phi(x_t) + w_t, \forall t \qquad (13b)$$

where $x_t \in \mathbb{R}^n$ is the state of the system, while $w_t$ is a i.i.d. random disturbance and $x_0$ is given initial state. To solve this problem we use Approximate Dynamic Programming (ADP) (Wang and Boyd 2010) that solves a proxy quadratic problem

$$\min_{u_t} u_t^T P u_t + x_t Q u_t + q^t u_t \qquad \text{s.t. } \|u_t\|_2 \leq 1 \quad (14)$$

We can use the optimization layer as shown in Fig.2(a) and update the problem variables (e.g. $P, Q, q$) using gradient descent. We use the linear quadratic regulator (LQR) solution as initial solution (Kalman 1964). The optimization module is replicated for each time step $t$, similarly to Recursive Neural Network (RNN).

We can build a resilient version of the controller in the hypothesis that an adversarial is able to inject a noise of limited energy, but arbitrary dependent on the control $u$, by solving the following bilevel optimization problem

$$\max_{\epsilon} \ Q(u_t, x_t + \epsilon) \qquad \text{s.t. } \|\epsilon\| \leq \sigma \qquad (15a)$$

$$u_t(\epsilon) = \arg\min_{u_t} Q(u_t, x_t) \qquad \text{s.t. } \|u_t\|_2 \leq 1 \qquad (15b)$$

where $Q(u, x) = u^T P u + x_t Q u + q^t u$ and we want to learn the parameters $z = (P, Q, q)$, where $y = u_t$, $x = \epsilon$ of Eq.1.

| gradient type | accuracy [12x12 maps] | | accuracy [18x18 maps] | | accuracy [24x24 maps] | |
|---|---|---|---|---|---|---|
| | train | validation | train | validation | train | validation |
| BiGrad(BB) | $95.8 \pm 0.2$ | $\mathbf{94.5} \pm 0.2$ | $\mathbf{97.1} \pm 0.0$ | $\mathbf{96.4} \pm 0.2$ | $98.0 \pm 0.0$ | $\mathbf{97.8} \pm 0.0$ |
| BiGrad(PT) | $91.7 \pm 0.1$ | $91.6 \pm 0.1$ | $94.3 \pm 0.0$ | $94.2 \pm 0.1$ | $95.7 \pm 0.0$ | $95.6 \pm 0.1$ |
| BB-1 | $95.9 \pm 0.2$ | $91.7 \pm 0.1$ | $96.7 \pm 0.2$ | $94.5 \pm 0.1$ | $97.1 \pm 0.1$ | $96.3 \pm 0.2$ |
| PT-1 | $88.3 \pm 0.2$ | $87.5 \pm 0.2$ | $90.9 \pm 0.4$ | $90.6 \pm 0.5$ | $92.8 \pm 0.1$ | $92.8 \pm 0.2$ |
| SL | $\mathbf{100.0} \pm 0.0$ | $26.2 \pm 2.4$ | $\mathbf{99.9} \pm 0.1$ | $20.2 \pm 0.5$ | $\mathbf{99.1} \pm 0.2$ | $14.0 \pm 1.0$ |

Table 2: Performance on the Dynamic Programming Problem with Interdiction. SL uses ResNet18.

| $L_\infty \leq \alpha$ | DCNN | Bi-DCNN | CNN | CNN* |
|---|---|---|---|---|
| 0 | $62.9 \pm 0.3$ | $\mathbf{64.0} \pm 0.4$ | $63.4 \pm 0.7$ | $63.6 \pm 0.5$ |
| 5 | $42.6 \pm 1.0$ | $\mathbf{44.5} \pm 0.2$ | $43.8 \pm 1.2$ | $44.3 \pm 1.0$ |
| 10 | $23.5 \pm 1.5$ | $\mathbf{25.3} \pm 0.8$ | $24.3 \pm 1.0$ | $24.2 \pm 1.0$ |
| 15 | $14.4 \pm 1.4$ | $\mathbf{15.6} \pm 0.7$ | $14.6 \pm 0.7$ | $14.3 \pm 0.4$ |
| 20 | $9.1 \pm 1.2$ | $\mathbf{10.0} \pm 0.6$ | $9.2 \pm 0.4$ | $8.9 \pm 0.2$ |
| 25 | $6.1 \pm 1.0$ | $\mathbf{6.8} \pm 0.5$ | $6.0 \pm 0.2$ | $5.9 \pm 0.2$ |
| 30 | $3.9 \pm 0.7$ | $\mathbf{4.4} \pm 0.5$ | $3.9 \pm 0.2$ | $3.9 \pm 0.1$ |

Table 3: Performance on the adversarial attack with discrete features, with $Q = 10$. DCNN is the single level discrete CNN, Bi-DCNN is the bilevel discrete CNN, CNN is the vanilla CNN, while CNN* is the CNN where we add the bilevel discrete layer after vanilla training.

We evaluate the performance to verify the viability of the proposed approach and compare with LQR and OptNet (Amos and Kolter 2017), where the outer problem is substituted with a best response function that computes the adversarial noise based on the computed output; in this case the adversarial noise is a scaled version of $Qu$ of Eq.14. Tab.1 and Fig.2(b) present the performance using BiGrad, LQR and the adversarial version of OptNet. BiGrad improves over two-step OptNet (Tab.1), because is able to better model the interaction between noise and control dynamic.

## 5.2 Adversarial ML with discrete latent variables

Machine learning models are heavily affected by the injection of intentional noise (Madry et al. 2017; Goodfellow, Shlens, and Szegedy 2014). Adversarial attack typically requires the access to the machine learning model, in this way the attack model can be used during training to include its effect. Instead of training an end-to-end system as in (Goldblum, Fowl, and Goldstein 2019), where the attacker is aware of the model, we consider the case where the attacker can inject a noise at feature level, as opposed at input level (as in (Goldblum, Fowl, and Goldstein 2019)), this allows us to model the interaction as a bilevel problem. Thus, to demonstrate the use of a bilevel layer, we design a system that is composed of a feature extraction layer, followed by a discretization layer that operates on the space of $\{0,1\}^m$, where $m$ is the hidden feature size, followed by a classification layer. The network used in the experiments is composed of two convolutional layers with max-pooling and two linear layers, all with relu activation functions, while the classification is a linear layer. We consider an more limited attacker that is not aware of the loss function of the model and does not have access to the full model, but rather only to the input of the discrete layer and is able two switch $Q$ discrete variables, The interaction of the discrete layer with the attacker is described by the following bilevel problem:

$$\min_{x \in Q} \max_{y \in B} \langle z + x, y \rangle. \tag{16}$$

where $Q$ represents the sets of all possible attack, $B$ the budget of the discritization layer and $y$ is the output of the layer. For the simulation, we compute the solution by sorting the features by values and considering only the first B values, while the attacker will obscure (i.e. set to zero) the first $Q$ positions. The output $y$ thus will have ones on the $Q$ to $B$ non-zero positions, and zero elsewhere. We train three models, on CIFAR-10 dataset for 50 epochs. For comparison we consider:1) the vanilla CNN network (i.e. without the discrete features); 2) the network with the single level problem (i.e. the single-level problem without attacker) and; 3) the network with the bilevel problem (i.e. the min-max discretization problem defined in Eq.16). We then test the networks to adversarial attack using the PGD (Madry et al. 2017) attack similar to (Goldblum, Fowl, and Goldstein 2019). Similar results apply for FGSM attack (Fast Gradient Sign Attack) (Goodfellow, Shlens, and Szegedy 2014). We also tested the network trained as vanilla network, where we added the min-max layer after training. From the results (Tab.3), we notice: 1) The min-max network shows improved resilience to adversarial attack wrt to the vanilla network, but also with respect to the max (single-level) network; 2) The min-max layer applied to the vanilla trained network is beneficial to adversarial attack; 3) The min-max network does not significantly change performance in presence of adversarial attack at the discrete layer (i.e. between Q=0 and Q=10). This example shows how bilevel-layers can be successfully integrated into Machine Learning system as differentiable layers.

## 5.3 Dynamic Programming: Shortest path with Interdiction

We consider the problem of Shortest Path with Interdiction, where the set of possible valid paths (see Fig.3(a)) is $Y$ and the set of all possible interdiction is $X$. The mathematical problem can be written as

$$\min_{y \in Y} \max_{x \in X} \langle z + x \odot w, y \rangle \tag{17}$$

where $\odot$ is the element wise product.This problem is multi-linear in the discrete variables $x, y, z$. The $z, w$ variables are output of neural network whose input are the Warcraft II tile images. The aim is to train the parameters of weight

| gradient type | k | accuracy train | accuracy validation | k | accuracy train | accuracy validation | k | accuracy train | accuracy validation |
|---|---|---|---|---|---|---|---|---|---|
| BiGrad(BB) | 8 | $89.2 \pm 0.1$ | $89.4 \pm 0.2$ | 10 | $91.9 \pm 0.1$ | $\mathbf{92.0} \pm 0.1$ | 12 | $93.5 \pm 0.1$ | $93.5 \pm 0.2$ |
| BiGrad(PT) | 8 | $89.3 \pm 0.0$ | $\mathbf{89.4} \pm 0.1$ | 10 | $92.0 \pm 0.0$ | $91.9 \pm 0.1$ | 12 | $\mathbf{93.7} \pm 0.1$ | $\mathbf{93.7} \pm 0.1$ |
| BB-1 | 8 | $84.0 \pm 0.4$ | $83.9 \pm 0.4$ | 10 | $87.4 \pm 0.3$ | $87.5 \pm 0.4$ | 12 | $89.3 \pm 0.1$ | $89.3 \pm 0.1$ |
| PT-1 | 8 | $84.1 \pm 0.4$ | $84.1 \pm 0.3$ | 10 | $87.3 \pm 0.3$ | $87.0 \pm 0.3$ | 12 | $89.3 \pm 0.0$ | $89.5 \pm 0.2$ |
| SL | 8 | $\mathbf{94.2} \pm 5.0$ | $10.7 \pm 3.9$ | 10 | $\mathbf{92.7} \pm 5.4$ | $9.4 \pm 0.4$ | 12 | $91.4 \pm 2.3$ | $9.3 \pm 1.2$ |

Table 4: Performance in term of accuracy of the TSP use case with interdiction. SL has higher accuracy during train, but fails in at test time.

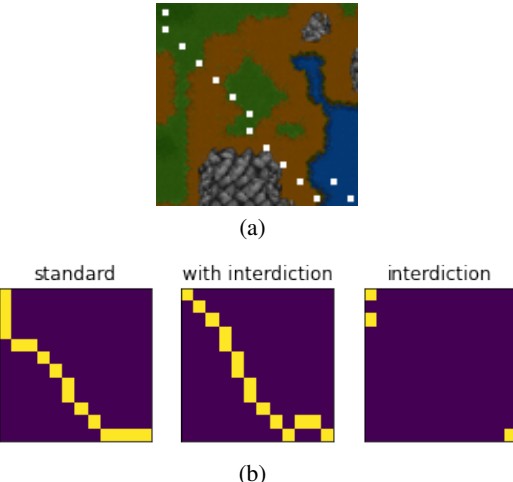

(a)

(b)

Figure 3: (a) Example Shortest Path in the Warcraft II tile set of (Guyomarch 2017). (b) Example Shortest Path without (left) and with interdiction (middle). Even a small interdiction (right) has a large effect on the output.

network, such that we can solve the shortest path problem only based on the input image. For the experiments, we followed and adapted the scenario of (Pogančić et al. 2019) and used the Warcraft II tile maps of (Guyomarch 2017). We implemented the interdiction Game using a two stage min-max-min algorithm (Kämmerling and Kurtz 2020). In Fig.3(b) it is possible to see the effect of interdiction on the final solution. Tab.2 shows the performances of the proposed approaches, where we allow for $B = 3$ interdictions and we used tile size of $12 \times 12$, $18 \times 18$, $24 \times 24$. The loss function is the Hamming and $\ell_1$ loss evaluated on both the shortest path $y$ and the intervention $x$. The gradient estimated using Eq.11 (BB) provides more accurate results, at double of computation cost of PT. Single level BB-1 approach outperforms PT, but shares similar computational complexity, while single level PT-1 is inferior to PT. As expected, SL outperforms other methods during training, but completely fails during validation. Bigrad improves over single-level approaches, because includes the interaction of the two problems.

## 5.4 Combinatorial Optimization: Travel Salesman Problem (TSP) with Interdiction

Travel Salesman Problem (TSP) with interdiction consists of finding shortest route $y \in Y$ that touches all cities, where some connections $x \in X$ can be removed. The mathematical

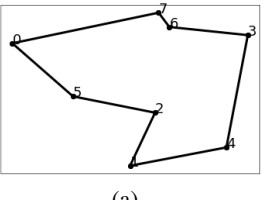
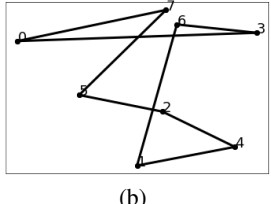

(a)          (b)

Figure 4: Example of TSP with $8$ cities and the comparison of a TSP tour without (a) or with (b) a single interdiction. Even a single interdiction has a large effect on the final tour.

problem to solve is given by

$$\min_{y \in Y} \max_{x \in X} \langle z + x \odot w, y \rangle \tag{18}$$

where $z, w$ are cost matrices for salesman and interceptor. Similar to the dynamic programming experiment, we implemented the interdiction Game using a two stage min-max-min algorithm (Kämmerling and Kurtz 2020). Fig.4 shows the effect of a single interdiction. The aim is to learn the weight matrices, trained with interdicted solution on subset of the cities. Tab.4 describes the performance in term of accuracy on both shortest tour and intervention. We use Hamming and $\ell_1$ loss function. We only allow for $B = 1$ intervention, but considered $k = 8, 10$ and $12$ cities from a total of $100$ cities. Single and two level approaches perform similarly in the train and validation. Since the number of interdiction is limited to one, the performance of the single level approach is not catastrophic, while the supervised learning approach completely fails in the validation set. Bigrad thus improves over single-level and SL approaches. Since Bigrad(PT) has similar performance of BiGrad(BB), thus PT is preferable in this scenario, since it requires less computation resources.

## 6 Conclusions

BiGrad generalizes existing single level gradient estimation approaches and is able to incorporate Bilevel Programming as learnable layer in modern machine learning frameworks, which allows to model conflicting objectives as in adversarial attack. The proposed novel gradient estimators are also efficient and the proposed framework is widely applicable to both continuous and discrete problems. The impact of Bi-Grad has a marginal or similar cost with respect to the complexity of computing the solution of the Bilevel Programming problems. We show how BiGrad is able to learn complex logic, when the cost functions are multi-linear.

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

# A   Supplementary Material; BiGrad: Differentiating through Bilevel Optimization Programming

## A.1   Relationship with other related work

**Predict then optimize**   Predict then Optimize (two-stage) (Elmachtoub and Grigas 2017; Ferber et al. 2020) or solving linear programs and submodular maximization from (Wilder, Dilkina, and Tambe 2019) solve optimization problems when the cost variable or the minimization function is directly observable. In contrary, in our approach we only have access to a loss function on the output of the bilevel problem, thus allowing to use as a layer.

## A.2   Proofs

*Proof of Linear Equality constraints.*   Here we show that

$$x(u) = x_0 + A^\perp u \tag{19}$$

includes all solution of $Ax = b$. First we have that $AA^\perp = 0$ and $Ax_0 = b$ by definition. This implies that $Ax(u) = A(x_0 + A^\perp u) = Ax_0 = b$. Thus $\forall u \to Ax(u) = b$. The difference $x' - x_0$ belongs to the null space of $A$, indeed $A(x' - x_0) = Ax' - Ax_0 = b - b = 0$. The null space of $A$ has size $n - \rho(A)$. If $\rho(A) = n$, where $A \in \mathbb{R}^{m \times n}, m \geq n$, then there is only one solution $x = x_0 = A^\dagger b$, $A^\dagger$ the pseudo inverse of $A$. If $\rho(A) < n$, then $\rho(A^\perp)) = n - \rho(A)$ is a based of all vectors s.t. $Ax(u) = b$, since $\rho(A^\perp)) = n - \rho(A)$ is the size of the null space of $A$. In fact $A^\perp$ is the base for the null space of $A$. The same applies for $y(v) = y_0 + B^\perp v$ and $By(v) = c$.   □

*Proof of Theorem 1.*   The second equation is derived by imposing the optimally condition on the inner problem. Since we do not have inequality and equality constraints we optimal solution shall equate the gradient w.r.t. $y$ to zero, thus $G = \nabla_y g = 0$. The first equation is also related to the optimality of the $x$ variable w.r.t. to the total derivative or hypergradient, thus we have that $0 = d_x f = \nabla_x f + \nabla_y f \nabla_x y$. In order to compute the variation of $y$, i.e. $\nabla_x y$ we apply the implicit theorem to the inner problem, i.e. $\nabla_x G + \nabla_y G \nabla_x y = 0$, thus obtaining $\nabla_x y = -\nabla_y^{-1} G \nabla_x G$.   □

*Proof of Theorem 2.*   In order to prove the theorem, we use the Discrete Adjoin Method (DAM). Let consider a cost function or functional $L(x, y, z)$ evaluated at the output of our system. Our system is defined by the two equations $F = 0, G = 0$ from Theorem 1. Let us first consider the total variations: $dL$, $dF = 0$, $dG = 0$, where the last conditions are true by definition of the bilevel problem. When we expand the total variations, we obtain

$$
\begin{aligned}
dL &= \nabla_x L dx + \nabla_y L dy + \nabla_z L dz \\
dF &= \nabla_x F dx + \nabla_y F dy + \nabla_z F dz \\
dG &= \nabla_x G dx + \nabla_y G dy + \nabla_z G dz
\end{aligned}
$$

We now consider $dL + dF\lambda + dG\gamma = [\nabla_x L + \nabla_x F\lambda + \nabla_x G\gamma]dx + [\nabla_y L + \nabla_y F\lambda + \nabla_y G\gamma]dy + [\nabla_z L + \nabla_z F\lambda +$

$\nabla_z G\gamma]dz$. We ask the first two terms to be zero to find the two free variables $\lambda, \gamma$:

$$
\begin{aligned}
\nabla_x L + \nabla_x F\lambda + \nabla_x G\gamma &= 0 \tag{20} \\
\nabla_y L + \nabla_y F\lambda + \nabla_y G\gamma &= 0 \tag{21}
\end{aligned}
$$

or in matrix form

$$
\begin{vmatrix} \nabla_x F & \nabla_x G \\ \nabla_y F & \nabla_y F \end{vmatrix} \begin{vmatrix} \lambda \\ \gamma \end{vmatrix} = - \begin{vmatrix} \nabla_x L \\ \nabla_y L \end{vmatrix}
$$

We can now compute the $d_z L = \nabla_z L + \nabla_z F\lambda + \nabla_z G\gamma$ with $\lambda, \gamma$ from the previous equation.   □

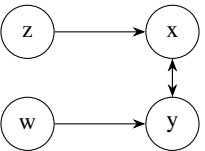

Figure 5: Discrete Bilevel Variables: Dependence diagram

*Proof of Theorem 3.*   The partial derivatives are obtained by using the perturbed discrete minimization problems defined by Eqs.24. We first notice that $\nabla_x \min_{y \in Y} \langle x, y \rangle = \arg\min_{y \in Y} \langle x, y \rangle$. This result is obtained by the fact that $\min_{y \in Y} \langle x, y \rangle = \langle x, y^* \rangle$, where $y^* = \arg\min_{y \in Y} \langle x, y \rangle$ and applying the gradient w.r.t. the continuous variable $x$; while Eqs. 23 are the expected functions of the perturbed minimization problems. Thus, if we compute the gradient of the perturbed minimizer, we obtain the optimal solution, proper scaled by the inner product matrix. For example $\nabla_x \tilde{\Phi}_\eta = Ax^*(z, y)$, with $A$ the inner product matrix. To compute the variation on the two parameter variables, we have that $dL = \nabla_x L dx + \nabla_y L dy + \nabla_z L dz + \nabla_w L dw$ and that $dw/dz = 0, dz/dw = 0$ from the dependence diagram of Fig.5   □

## A.3   Gradient Estimation based on perturbation

We can use the gradient estimator using the perturbation approach proposed in (Berthet et al. 2020). We thus have

$$
\begin{aligned}
\nabla_z x(z, y) &= A^{-1} \nabla_{z^2}^2 \tilde{\Phi}_\eta(z, y) \mid_{\eta \to 0} \tag{22a} \\
\nabla_w y(w, z) &= C^{-1} \nabla_{w^2}^2 \tilde{\Psi}_\eta(w, z) \mid_{\eta \to 0} \tag{22b} \\
\nabla_x y(x, w) &= D^{-1} \nabla_{x^2}^2 \tilde{\Theta}_\eta(x, w) \mid_{\eta \to 0} \tag{22c} \\
\nabla_y x(z, y) &= B^{-1} \nabla_{y^2}^2 \tilde{W}_\eta(z, y) \mid_{\eta \to 0} \tag{22d} \\
\nabla_z y &= \nabla_x y \nabla_z x \tag{22e}
\end{aligned}
$$

and

$$
\begin{aligned}
\tilde{\Phi}_\eta(z, y) &= \mathbb{E}_{u \sim U} \Phi(z + \eta u, y) \tag{23a} \\
\tilde{\Psi}_\eta(w, x) &= \mathbb{E}_{u \sim U} \Psi(w + \eta u, x) \tag{23b} \\
\tilde{\Theta}_\eta(x, w) &= \mathbb{E}_{u \sim U} \Psi(w, x + \eta u) \tag{23c} \\
\tilde{W}_\eta(y, z) &= \mathbb{E}_{u \sim U} \Phi(z, y + \eta u) \tag{23d}
\end{aligned}
$$

, while

$$\Phi(z, y) = \min_{x \in X} \langle z, x \rangle_A + \langle y, x \rangle_B \quad (24a)$$

$$\Psi(w, x) = \min_{y \in Y} \langle w, y \rangle_C + \langle x, y \rangle_D \quad (24b)$$

which are valid under the conditions of (Berthet et al. 2020), while $\tau$ and $\mu$ are hyper-parameters.

## A.4 Alternative derivation

Let consider the problem $\min_{x \in K} \langle z, x \rangle_A$ and let us define $\Omega_x$ a penalty term that ensures $x \in K$. We can define the generalized lagragian $\mathbb{L}(z, x, \Omega) = \langle z, x \rangle_A + \Omega_x$. One example of $\Omega_x = \lambda^T |x - K(x)|$ or $\Omega_x = -\ln|x - K(x)|$ where $K(x)$ is the projection into $K$. To solve the Lagragian, we solve the unconstrained problem $\min_x \max_{\Omega_x} \mathbb{L}(z, x, \Omega_x)$. At the optimal point $\nabla_x \mathbb{L} = 0$. Let us define $F = \nabla_x \mathbb{L} = A^T z + \Omega'_x$, then $\nabla_x F = \Omega''_x$ and $\nabla_z F = A^T$. If we have $F(x, z) = 0$ and a cost function $L(x, z)$, we can compute $\mathrm{d}_z L = \nabla_z L - \nabla_x L \nabla_x^{-1} F \nabla_z F$. Now $F(x, z, \Omega_x) = 0$, we can apply the previous result and $\mathrm{d}_z L = \nabla_z L - \nabla_x L \Omega_x''^{-1} A^T$. If we assume $\Omega''_x = I$ and $\nabla_z L = 0$, then $\mathrm{d}_z L = -A \nabla_x L$.

## A.5 Memory Efficiency

For continuous optimization programming, by separating the computation of the solution and the computation of the gradient around the current solution we 1) compute the gradient more efficiently, in particular we compute second order gradient taking advantage of the vector-jacobian product (push-back operator) formulation without explicitly inverting and thus building the jacobian or hessian matrices; 2) use more advanced and not differentialble solution techniques to solve the bilevel optimization problem that would be difficult to integrate using automatic differentiable operations. Using VJP we reduce memory use from $O(n^2)$ to $O(n)$. Indeed using an iterative solver, like generalized minimal residual method (GMRES) (Saad and Schultz 1986), we only need to evaluate the gradients of Eq.5 and not invert the matrix neither materialize the large matrix and computing matrix-vector products. Similarly, we use Conjugate Gradient (CG) method to compute Eq.4, which requires to only evaluate the gradient at the current solution and nor inverting neither materializing the Jacobian matrix. An imple-

mentation of a bilevel solver would have a memory complexity of $O(Tn)$, where $T$ are the number of iterations of the bilevel algorithm.

## A.6 Experimental Setup and Computational Resources

For the Optimal Control with adversarial disturbance we follow a similar setup of (Agrawal et al. 2019a), where we added the adversarial noise as described in the experiments. For the Combinatorial Optimization, we follow the setup of (Pogančić et al. 2019). The dataset is generated by solving the bilevel problem on the same data of (Pogančić et al. 2019). For section 5.3, we use the warcraft terrain tiles and generate optimal bilevel solution with the correct parameters $(z, w)$, where $z$ is the terrain transit cost and $w$ is the interdiction cost, considered constant to 1 in our experiment. $X$ is the set of all feasible interdictions, in our experiment we allow the maximum number of interdictions to be $B$. For section 5.4, on the other hand the $z$ represents the true distances among cities and $w$ a matrix of the interdiction cost, both unknown to the model. $X$ is the set of all possible interdictions. In these experiments, we solved the bilevel problem using the min-max-min algorithm (Kämmerling and Kurtz 2020). For the Adversarial Attack, we used two convolutional layers with max-pooling, relu activation layer, followed by the discrete layer of size $m = 2024$, $B = 100$, $Q = 0, 10$. A final linear classification layer is used to classify CIFAR10. We run over 3 runs, 50 epochs, learning rate $lr = 3e - 4$ and Adam optimizer. Experiments were conducted using a standard server with 8 CPU, 64Gb of RAM and GeForce RTX 2080 GPU with 6Gb of RAM.

## A.7 Jacobian-Vector and Vector-Jacobian Products

The Jacobian-Vector Product (JVP) is the operation that computes the directional derivative $J_f(x)u$, with direction $u \in \mathbb{R}^m$, of the multi-dimensional operator $f : \mathbb{R}^m \to \mathbb{R}^n$, with respect to $x \in \mathbb{R}^m$, where $J_f(x)$ is the Jacobian of $f$ evaluated at $x$. On the other hand, the Vector-Jacobian product (VJP) operation, with direction $v \in \mathbb{R}^n$, computes the adjoint directional derivative $v^T J_f(x)$. JVP and VJP are the essential ingredient for automatic differentiation (Elliott 2018; Baydin et al. 2018).