# OpenReview forum: "BiGrad: Differentiating through Bilevel Optimization Programming"
_AAAI.org/2022/Workshop/AdvML — AAAI-22 AdvML Workshop LongPaper_

### Official Review · Reviewer_hDgc · 2021-11-30
**A novel method for diferentiating a bilevel programming solver**

**Rating:** 7
**Confidence:** 4

**Review:**

This paper proposed a novel method for calculating gradients of a bilevel programming solver. This problem has a wide range of applications like adversarial training, GAN, and combinatorial optimization problems.

Pros
The method is novel and interesting and it can easily be embedded into a neural network as a layer. The author also implements it in several examples like linear and non-linear inequality constraints.

Cons
I think the author could provide more examples or applications about adversarial learning. For example, adversarial training (AT) is also a bilevel optimization problem. Can your method be used in AT?

---

### Decision · Program_Chairs · 2021-12-01

**Decision:**

Accept (Long Paper)

**Comment:**

The reviewer agrees to accept this paper. Please address the reviewer's comment in the camera-ready version.